# Characterized *cis*-Fe$^V$(O)(OH) intermediate mimics enzymatic oxidations in the gas phase

Margarida Borrell[1], Erik Andris[2], Rafael Navrátil[2], Jana Roithová[2,3] & Miquel Costas [1]

Fe$^V$(O)(OH) species have long been proposed to play a key role in a wide range of biomimetic and enzymatic oxidations, including as intermediates in arene dihydroxylation catalyzed by Rieske oxygenases. However, the inability to accumulate these intermediates in solution has thus far prevented their spectroscopic and chemical characterization. Thus, we use gas-phase ion spectroscopy and reactivity analysis to characterize the highly reactive [Fe$^V$(O)(OH)($^{5tips3}$tpa)]$^{2+}$ (**3$^{2+}$**) complex. The results show that **3$^{2+}$** hydroxylates C–H bonds via a rebound mechanism involving two different ligands at the Fe center and dihydroxylates olefins and arenes. Hence, this study provides a direct evidence of Fe$^V$(O)(OH) species in non-heme iron catalysis. Furthermore, the reactivity of **3$^{2+}$** accounts for the unique behavior of Rieske oxygenases. The use of gas-phase ion characterization allows us to address issues related to highly reactive intermediates that other methods are unable to solve in the context of catalysis and enzymology.

[1] Institut de Quimica Computacional I Catalisi and Departament de Química, Universitat de Girona. Facultat de Ciències, Campus de Montilivi, 17071 Girona, Spain. [2] Department of Organic Chemistry, Faculty of Science, Charles University, Hlavova 2030/8128 43 Prague 2Czech Republic. [3] Institute for Molecules and Materials, Radboud University Nijmegen, Heyendaalseweg 135, 6525 AJ Nijmegen, The Netherlands. These authors contributed equally: Margarida Borrell, Erik Andris. Correspondence and requests for materials should be addressed to J.R. (email: j.roithova@science.ru.nl) or to M.C. (email: miquel.costas@udg.edu)

High-valent iron species are highly reactive molecules involved in numerous oxidative processes of synthetic and biological relevance[1,2]. In particular, Fe(V) intermediates have been proposed as the oxidation agents in key organic synthesis reactions, such as C–H, C=C and arene oxidations, and in energy-related transformations, including water oxidation[3]. Moreover, iron-dependent enzymes such as cytochrome P450 and Rieske oxygenases presumably use formal Fe(V) intermediates to oxidize inert substrates, including alkanes or arenes. Cytochrome P450 has been shown to use an oxoiron(IV) porphyrin cation radical intermediate termed compound I (cpd I) in C–H oxidation reactions[4], whereas Rieske oxygenases may use a non-detectable oxoiron(V) intermediate in the syn-dihydroxylation of arenes and in metabolic C–H oxidations, although no direct evidence has been reported thus far[5–7]. Oxoiron(V) complexes are extremely challenging targets for synthetic inorganic chemistry because of their high reactivity. Accordingly, no crystal structure is available, and spectroscopically characterized examples remain exceedingly rare[8–13].

Inspired by iron oxygenases, chemists have intensively exploited iron coordination complexes as catalysts, also thanks to the availability of this metal[14]. Complexes with tetradentate aminopyridine ligands are particularly interesting because they can use hydrogen peroxide to catalyze enzyme-like stereoretentive C–H and C=C oxidations (Fig. 1a)[15–17]. Extensive mechanistic studies based on product analysis, isotopic labeling and computations have indirectly shown that these complexes operate via Fe$^V$(O)(X) (X = alkyl carboxylate or OH) reactive species[18–21].

Carboxylic acids assist the heterolytic cleavage of the O–O bond (Fig. 1b), forming a reactive Fe$^V$(O)(O$_2$CR) (R = alkyl) intermediate (IIIb in Fig. 1b) that epoxidizes olefins and hydroxylates alkanes[20,22]. Quite recently, this Fe$^V$(O)(O$_2$CR) intermediate was accumulated using a robust ligand frame, enabling its spectroscopic and chemical characterization

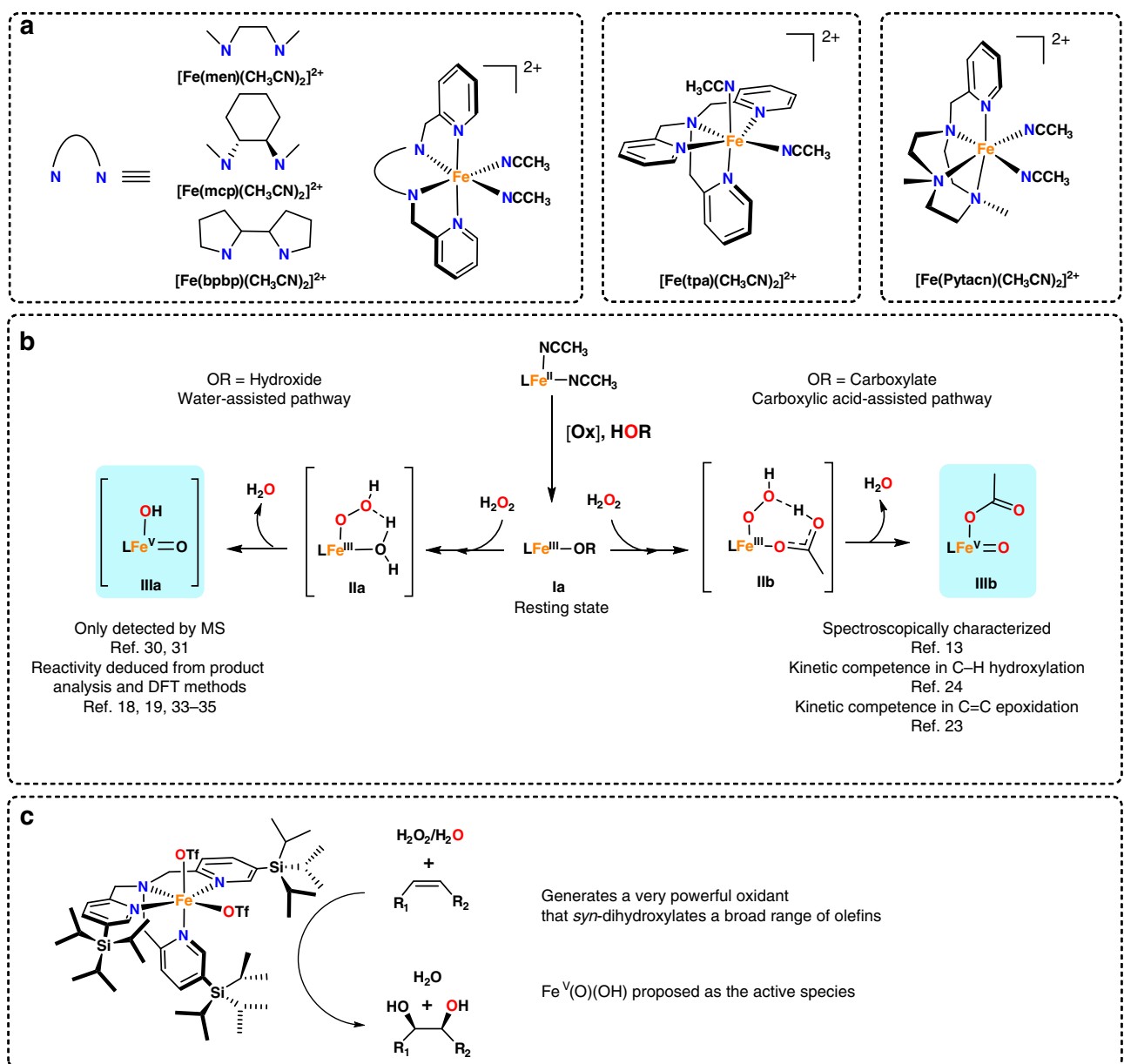

**Fig. 1** Mechanistic frame of iron-catalyzed oxidations with tetradentate aminopyridine ligands. **a** Iron catalysts based on tetradentate aminopyridine ligands that catalyze stereoretentive C–H and C=C oxidations. **b** Mechanistic frame for the generation of Fe(V) species in the presence (right) or absence (left) of carboxylic acids. **c** Schematic diagram of [Fe$^{II}$(CF$_3$SO$_3$)$_2$($^{5tips3}$tpa)] (**1**) and its catalytic oxidation activity

and, therefore, providing a solid foundation for the mechanistic proposal[13,23,24].

In the absence of carboxylic acids, O–O cleavage may be assisted by a water molecule, forming an $Fe^V(O)(OH)$ (**IIIa** in Fig. 1b) intermediate that hydroxylates alkanes and engages in the *syn*-dihydroxylations of olefins[25]. These $Fe^V(O)(OH)$ species have also been proposed to oxidize the water molecule under low pH condition[3]. Moreover, no $Fe^V(O)(OH)$ species accumulate in solution, consistently with their high reactivity.

The complex [Fe^II(CF_3SO_3)_2(5tips3tpa)], **1** (Fig. 1c, 5tips3tpa = *tris*(5-(triisopropyl)silyl-2-methylpyridyl)amine is a remarkable example of a catalyst operating through $Fe^V(O)(OH)$ intermediates. It efficiently catalyzes the *syn*-dihydroxylation of various olefins with $H_2O_2$[26,27] and is, thus, a sustainable alternative to traditional, Os- and Ru-based *syn*-dihydroxylating agents[28,29]. In addition, this catalyst shows outstanding selectivity properties; for example, olefins are highly chemoselectively *syn*-dihydroxylated while epoxidation is largely minimized. Moreover, electron-deficient olefins, and arenes, unreactive to Os-based reagents, are instantaneously dihydroxylated, thus indicating the involvement of extraordinarily powerful oxidizing species.

Before the present study, $Fe^V(O)(OH)$ species have been detected only by mass spectrometry (MS), and their formulation was derived from experiments using isotopically labeled reagents ($H_2^{18}O$ and $H_2^{18}O_2$)[30,31]. Although their reactivity has been inferred from MS analysis of catalytic reaction mixtures, their spectroscopic characterization and direct assessment of their reactivity has not been performed yet.

Herein, we spectroscopically characterized the proposed [Fe^V(O)(OH)(5tips3tpa)]^{2+} reactive intermediate in the gas phase by helium tagging infrared photodissociation (IRPD) spectroscopy[32]. We conclusively identify the terminal $Fe^V$=O and $Fe^V$–OH stretching vibrations of the Fe(O)(OH) unit. Furthermore, we confirm that [Fe^V(O)(OH)(5tips3tpa)]^{2+} hydroxylates C–H bonds in a rebound mechanism and performs the *syn*-dihydroxylation of alkenes and arenes. These reactions, previously described in enzymes and bioinspired oxidation catalysts, have only been previously understood based on product analysis and computational methods[5,6,19,30,33–35].

Thus, the present study reports the experimental characterization of the $Fe^V(O)(OH)$ species and demonstrates its chemical competence in bioinspired reactions, particularly in reactions relevant to Rieske oxygenases.

## Results

**Generation and ion-spectroscopy characterization of intermediates.** The reaction of **1** (0.4 mM) with $H_2O_2$ (10 equiv.) in acetonitrile at −40 °C, monitored by ultraviolet-visible (UV–vis) spectroscopy, produces a metastable purple species **2** ($\lambda_{max} = 544$ nm, $\varepsilon = 1300$ M$^{-1}$ cm$^{-1}$) (Fig. 2 and Supplementary Fig. 2). After **2** was formed in acetonitrile solution, the reaction mixture was analyzed by electrospray ionization mass spectrometry (ESI-

MS). Two peaks at $m/z = 444$ and 424 stand out in the ESI-MS spectrum (Supplementary Fig. 1). The former corresponds to the expected dicationic species [Fe^III(OOH)(CH_3CN)(5tips3tpa)]^{2+} (**2^{2+}**)[25], whereas the latter can be tentatively formulated as either [Fe^III(OOH)(5tips3tpa)]^{2+} (**2a^{2+}**) or [Fe(O)(OH)(5tips3tpa)]^{2+} (**3^{2+}**), wherein the O–O bond has been broken. Using helium-tagging IRPD spectroscopy[32] we were able to measure IR spectra of the mass-selected ions with $m/z$ 424 generated by electrospray ionization from the solution of **2** and ascertain that these indeed correspond to [Fe^V(O)(OH)(5tips3tpa)]^{2+} (**3^{2+}**).

We measured the IRPD spectrum of the ions (corresponding to the iron(V) intermediate **3^{2+}**) generated from the reaction mixture of **1** and $H_2^{16}O_2$. The spectrum was assigned by comparison with the spectra of isotopically labeled ions resulting from the oxidation of **1** with $H_2^{18}O_2$, $H_2^{16}O^{18}O$, and $D_2^{16}O_2$ (the mass-selected complexes contained the $^{56}Fe$ isotope if not mentioned otherwise, Fig. 3). We also analyzed the spectrum of naturally occurring $^{54}Fe$-labeled ions (**3^{2+}**($^{54}Fe$), Supplementary Fig. 3). The IRPD spectrum of [Fe^V(^{16}O,^{16}OH)(5tips3tpa)]^{2+} (**3^{2+}**, $m/z$ 424) (Fig. 3a, black) shows bands at 827 cm$^{-1}$ and 638 cm$^{-1}$ that shift to 797 cm$^{-1}$ and 616 cm$^{-1}$, respectively, upon double $^{18}O$ labeling [Fe^V(^{18}O,^{18}OH)(5tips3tpa)]^{2+}, (**3^{2+}**($^{18}O^{18}O$), $m/z$ 426, Fig. 3a). These bands can be interpreted as either Fe=O and Fe–OH stretching vibrations[9,13,36,37] or as O–O stretching and O–O–H bending vibrations (see the comparison with the theoretically predicted IR spectra in Fig. 3c)[38]. The frequencies of the characteristic $n$(Fe–O) and $n$(O–O) bands of [Fe^III(OOH)(tpa)(S)]^{2+} (S = solvent) have been determined by resonance Raman to be 626 cm$^{-1}$ and 789 cm$^{-1}$[38], respectively, which may be considered in reasonable agreement with those observed for **3**. To differentiate {Fe(O)(OH)} and {Fe(OOH)} binding motifs, we measured the IRPD spectrum of singly $^{18}O$ labeled ions [Fe^V(16/18O,18/16OH)(5tips3tpa)]^{2+} (**3^{2+}**($^{16}O^{18}O$), $m/z$ 425). If the 827 cm$^{-1}$ band would correspond to the O–O stretching mode, the band should redshift (the bond would be always labeled by $^{18}O$). Conversely, if the band would correspond to the Fe=O stretch, only half the band should disappear (the Fe=O bond is $^{18}O$ labeled in only 50% ions). Indeed, the second variant is observed in the experiment (Fig. 3b). This result allows us to assign the IRPD spectra to the [Fe^V(O)(OH)(5tips3tpa)]^{2+} intermediates. Fully consistent with this interpretation, the vibrational spectrum of **3^{2+}**($^{54}Fe$) ($m/z = 423$) shows that both bands are blueshifted by 3 cm$^{-1}$, as expected for spectral shifts of Fe–O bonds (Supplementary Fig. 3). Interestingly, Hooke's Law analysis of a Fe–O oscillator predicts shifts of 3 cm$^{-1}$ for both vibrations

The vibrational features of **3^{2+}** agree well with the DFT spectra of the [Fe^V(O)(OH)(5tips3tpa)]^{2+} complex with the S = 3/2 ground state as predicted at the B3LYP-D3/def2TZVP level (**^43^{2+}**, Fig. 3c, e). In addition to reproducing the experimentally determined energy and isotopic shifts of Fe=O and Fe–OH stretching vibrations, the computations predict a distinctive δ-OH

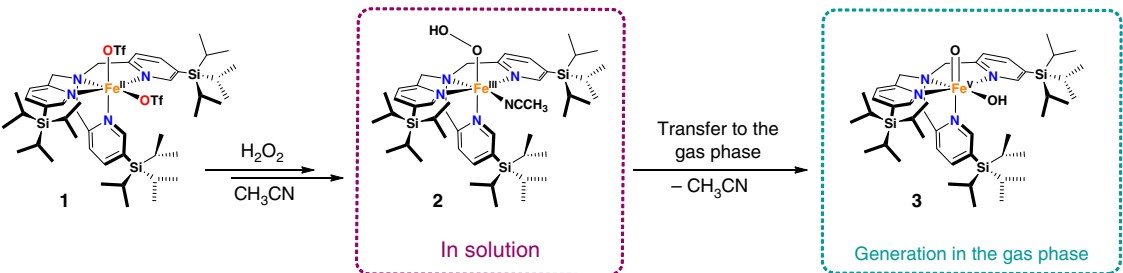

**Fig. 2** Generation of the iron(V) intermediate **3**. Schematic diagram of the formation of ferric hydroperoxide species **2** in solution and subsequent transfer of this species to the gas phase where the $Fe^V$ species **3** is generated

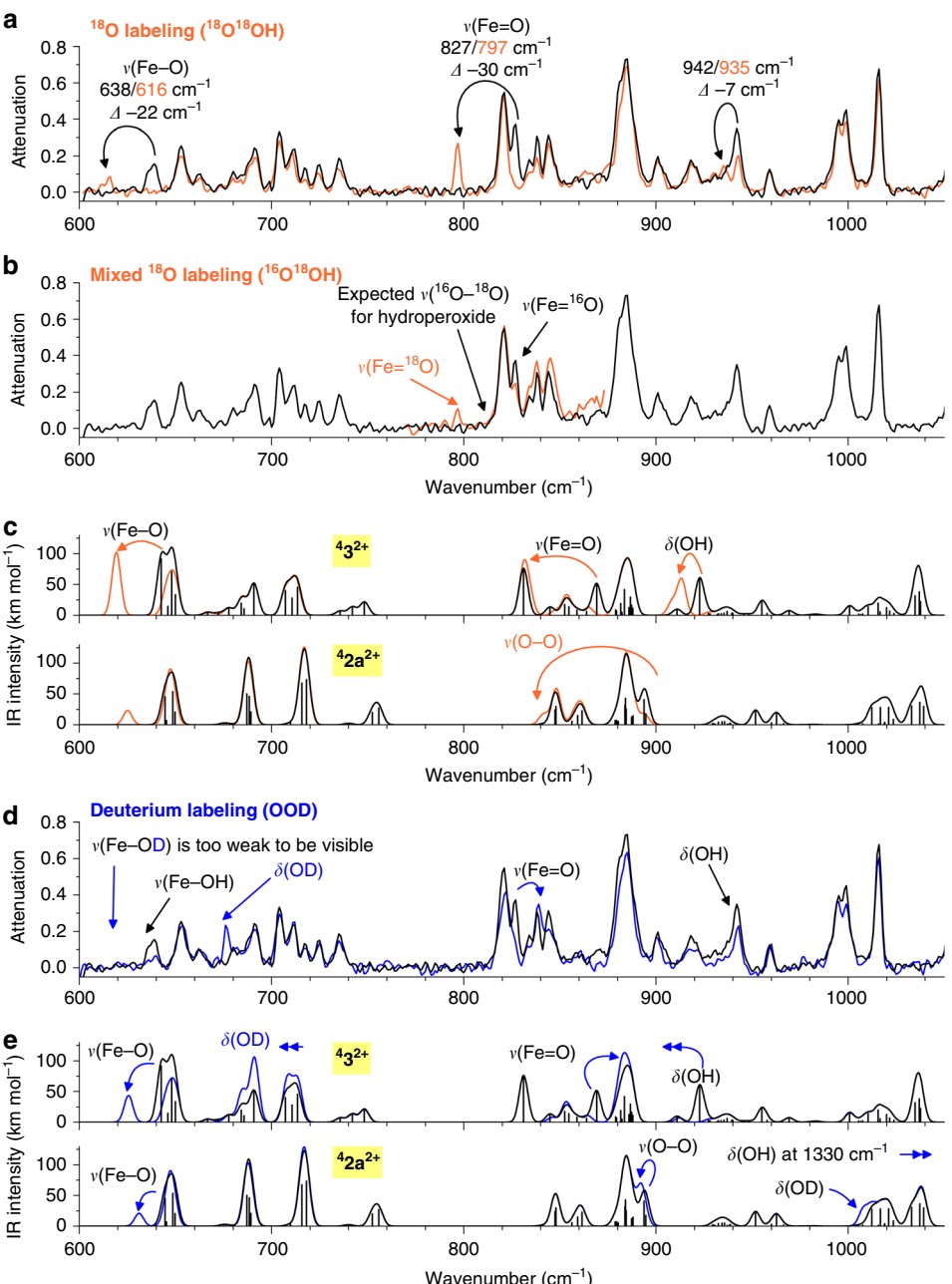

**Fig. 3** IRPD spectrum of the ions generated from the reaction mixture of **1** with $H_2O_2$ and their prediction. **a** IRPD spectra of $3^{2+}$ (black trace) and $3^{2+}(^{18}O^{18}O)$ (orange trace). **b** IRPD spectra of $3^{2+}$ and $3^{2+}(^{18}O^{16}O)$. **c** B3LYP-D3/def2TZVP predictions of the IR spectra for the $3^{2+}$ and $2a^{2+}$ complexes. **d** IRPD spectra of $3^{2+}$ and $3^{2+}(^{2}H)$. **e** B3LYP-D3/def2TZVP predictions of the IR spectra of $3^{2+}$ and $2a^{2+}$ complexes

vibration, sensitive to deuteration, and a blueshift of the Fe=O stretch in $3^{2+}(^{2}H)$. The blueshift of the Fe=O stretch in $3^{2+}(^{2}H)$ (Fig. 3d) is caused by coupling between the $v$(Fe=O) and $\delta$(OH) vibrations; this coupling is also evident when we calculate the IR spectrum of $3^{2+}$ labeled only at the Fe=O oxygen, wherein the $\delta$(OH) vibration redshifts (Supplementary Fig. 4e). We also considered the doublet state, but the calculations predict that this state is 12.3 kcal mol$^{-1}$ higher in energy than the quartet state. Its predicted IR spectrum is quite similar to that of the quartet state complex, except for a higher frequency of the Fe=O stretching vibration (Supplementary Fig. 4b).

The computed spectroscopic features of the $[Fe^{III}(OO(H/D)(^{5tips3}tpa)]^{2+}$ species were also considered. Interestingly, the computed O–O stretching frequency is basically insensitive to

deuteration of the hydroperoxide ligand, in line with the resonance Raman analysis of $[Fe^{III}(OO(H/D)(N4Py)]^{2+}$ [39], (N4Py = (1,1-di(pyridin-2-yl)-N,N-bis(pyridin-2-ylmethyl) methanamine)) and in sharp contrast with the shift of $[Fe^V(O)(OD)(^{5tips3}tpa)]^{2+}$. In conclusion, the vibrational spectra provide compelling evidence that $3^{2+}$ must be formulated as $[Fe^V(O)(OH)(^{5tips3}tpa)]^{2+}$, wherein the iron center is in the quartet state. This formulation actually reproduces the structure and spin ground state predicted by Siegbahn and Que for the parent $[Fe^V(O)(OH)(tpa)]^{2+}$, based on DFT calculations [40].

The isotopic composition of $3^{2+}$ shows that both oxygen atoms originate from a single $H_2O_2$ molecule, thus indicating that its formation is not assisted by a water molecule (Fig. 1b). In contrast, isotopic analyzes of diol products formed in catalytic

**Table 1 Structural and spectroscopic features of previously described Fe$^V$(O) complexes**

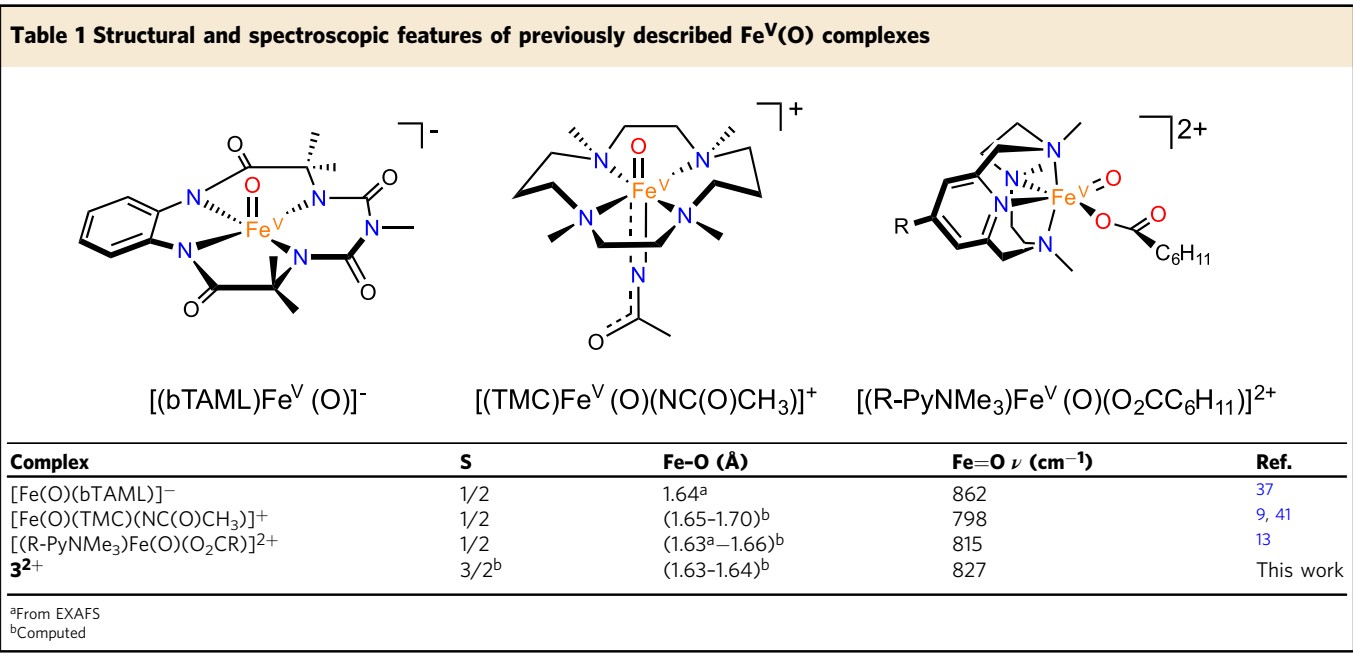

$[(bTAML)Fe^V(O)]^-$    $[(TMC)Fe^V(O)(NC(O)CH_3)]^+$    $[(R\text{-}PyNMe_3)Fe^V(O)(O_2CC_6H_{11})]^{2+}$

| Complex | S | Fe-O (Å) | Fe=O ν (cm$^{-1}$) | Ref. |
|---|---|---|---|---|
| $[Fe(O)(bTAML)]^-$ | 1/2 | 1.64$^a$ | 862 | 37 |
| $[Fe(O)(TMC)(NC(O)CH_3)]^+$ | 1/2 | (1.65–1.70)$^b$ | 798 | 9, 41 |
| $[(R\text{-}PyNMe_3)Fe(O)(O_2CR)]^{2+}$ | 1/2 | (1.63$^a$–1.66)$^b$ | 815 | 13 |
| $3^{2+}$ | 3/2$^b$ | (1.63–1.64)$^b$ | 827 | This work |

$^a$From EXAFS
$^b$Computed

olefin oxidation reactions conducted in acetonitrile show that $3^{2+}$ is formed in solution with the assistance of a water molecule, that is, $3^{2+}$ contains one oxygen atom from $H_2O_2$ and another from water[26]. Therefore, $3^{2+}$ must be formed by different mechanisms in solution and in gas phase. DFT calculations suggest that $3^{2+}$ is more than 6 kcal mol$^{-1}$ lower in energy than $2a^{2+}$ in the gas phase (see Supplementary Table 1). Therefore, elimination of acetonitrile from $2^{2+}$ in the gas phase may likely lead directly to the rearranged product $3^{2+}$.

The electronic spectrum of $3^{2+}$ could be also determined by photodissociation spectroscopy (Supplementary Fig. 5). The spectrum is characterized by two absorption bands at 440 nm and 530 nm, corresponding to a charge transfer transition, which is also well reproduced by TD-DFT calculations of $^43^{2+}$.

The complex $^43^{2+}$ is one of the few spectroscopically characterized Fe$^V$=O complexes thus far (Table 1) and the first example of an Fe$^V$(O)(OH) species. These species are frequently postulated as catalytic intermediates in iron-catalyzed biomimetic oxidations[16] and in the catalytic cycle of Rieske oxygenases[6]. In addition, $3^{2+}$ is also the single experimentally characterized example of an Fe(V) complex with a postulated S = 3/2 spin state. The energy of the Fe=O stretch in the doublet state complexes range from 798 cm$^{-1}$ to 862 cm$^{-1}$ [8,13]; thus, the energy of the Fe–O bond of $^43^{2+}$ falls within the same range. Moreover, Fe$^{IV}$=O complexes show Fe–O vibrations from 798 cm$^{-1}$ to 850 cm$^{-1}$ [2]. This suggests that the strength of the Fe=O bond is unaffected by electron removal from Fe$^{IV}$=O presumably because the electron is removed from non-bonding orbitals, with respect to the Fe=O bond (electronic configuration of S = 1 Fe$^{IV}$=O: $d_{xy}^2 d_{xz}^1 d_{yz}^1$; of S = 3/2 Fe$^V$=O: $d_{xy}^1 d_{xz}^1 d_{yz}^1$; Fe=O bond is along $z$-axis). Interestingly, the related iron(V) complex with oxo and acyloxo ligands had the doublet ground state (Table 1)[13]. According to our calculations, replacing of OH by CH$_3$COO in $3^{2+}$ would result in the spin change of the ground state to S = ½. This change is also associated with an energetic preference for the closed {LFe$^{III}$(OOCOCH$_3$)} form over the open, high valent {LFe$^V$(O)(OCOCH$_3$)} in the gas phase (see in the Supplementary Discussion and Supplementary Fig. 8)[42].

**Reactivity studies**. After establishing the structure of the iron(V) intermediate, we probed its reactivity with a series of substrates in collisional experiments in the gas phase[43,44]. We studied reactions of mass-selected ions where each ion interacted with only one molecule of a given reactant R. The detected ionic products are thus formed from a well-defined reactant complex [$^43^{2+}$·R] without involvement of any additional molecules such as water or another reactant molecule. The reactions of $^43^{2+}$ proceed efficiently, attesting its high reactivity. Remarkably, when the ion corresponding to [Fe$^{III}$(OOH)(CH$_3$CN)(5tips3tpa)]$^{2+}$ ($2^{2+}$, $m/z = 444.3$) was tested in similar experiments, no reactivity was observed (Supplementary Fig. 7). This lack of reactivity of the hydroperoxide species against organic molecules reproduces well the rather sluggish oxidant character of these species in solution[25,45].

The reaction of $^43^{2+}$($^2$H) with cyclohexene (Fig. 4a) yields two products. The dominant product ($m/z$ 465.3, 84%), an adduct between cyclohexene and $^43^{2+}$($^2$H), corresponds to a dihydroxylation reaction (Supplementary Table 2). The second ion product ($m/z = 416.3$, 16%) results from the oxygen transfer from $^43^{2+}$($^2$H) to olefin, most likely in an epoxidation reaction. Thus, these findings are in line with experimental results under catalytic oxidation conditions, thus showing that **1** catalyzes olefin oxidation, largely favoring *syn*-dihydroxylation over epoxidation reactions[26].

The reaction of $^43^{2+}$ with 1,3-cyclohexadiene also dominantly leads to the adduct resulting from the dihydroxylation of an olefinic site alongside with the oxygen transfer reaction (Supplementary Table 2). The addition reaction is accompanied in approximately 10% by subsequent water elimination probably driven by restoring the conjugated double bond system.

We rationalized the reaction pathways based on experiments with deuterated complex ([Fe$^V$(O)(OD)(5tips3tpa)]$^{2+}$ ($^43^{2+}$ ($^2$H), Fig. 4b). The deuterium atom allows us to follow the subsequent fragmentation pathways. The initially formed adduct complex (the dihydroxylation product) is long-lived and therefore allows for hydrogen scrambling (complex is isolated in the gas phase and does not interact with any other molecules/ions)[46,47]. Note that this complex is isolated in the gas phase, therefore contains the

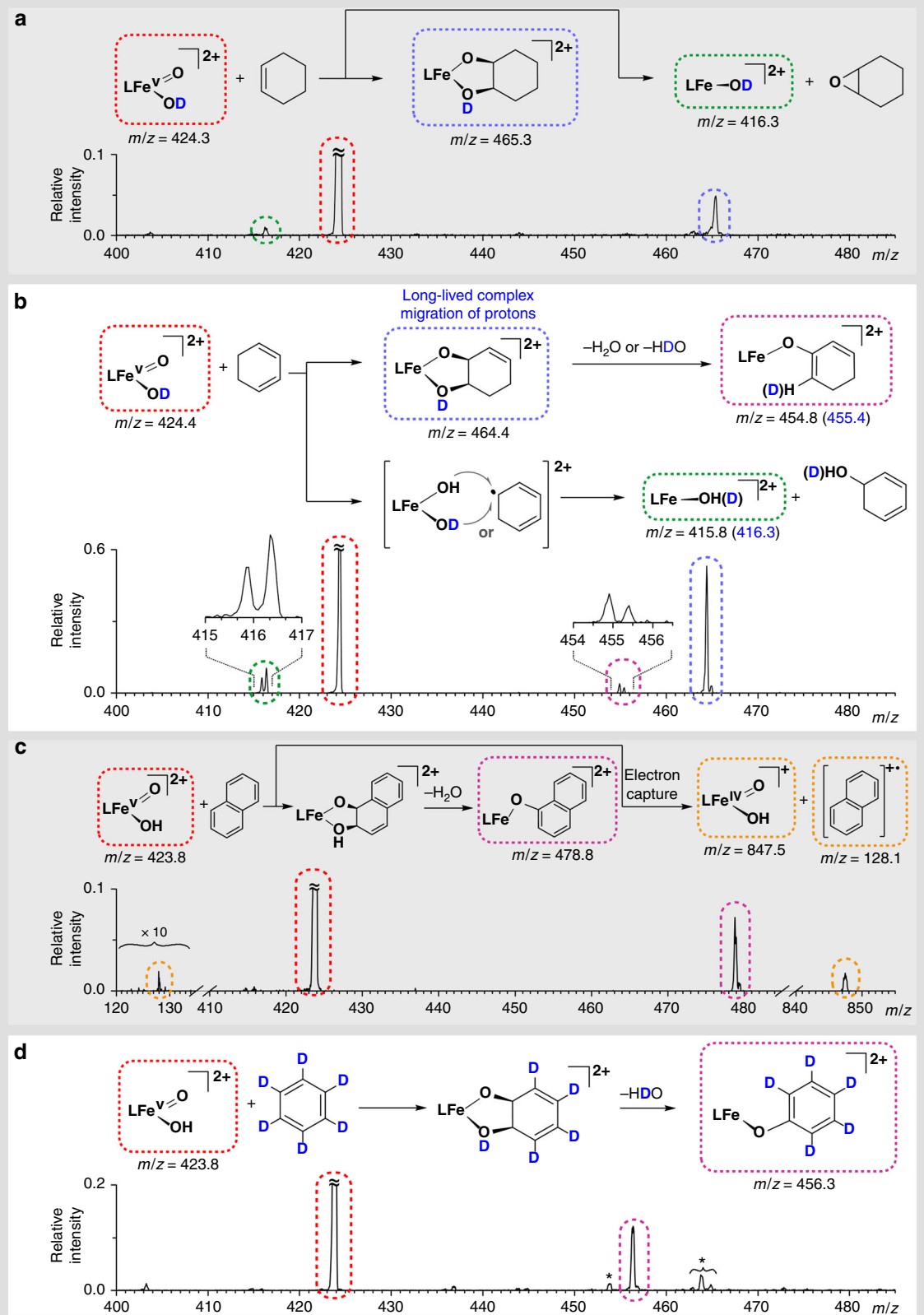

**Fig. 4** Ion-molecule reactivity of **3²⁺** and **3²⁺(²H)** in the gas phase. **a** 0.1 mTorr of cyclohexene, **b** 0.1 mTorr of 1,3-cyclohexadiene, **c** < 0.1 mTorr of naphthalene, and **d** 0.2 mTorr benzene (asterisks indicate impurities from previous measurements). All reactions were measured at nominally zero-collision energy determined from the retarding potential analysis

energy released by the exothermic interaction between the reactants and cannot dissipate this energy by interaction with other molecules. The subsequent dehydration of the adduct thus features as elimination of HDO or $H_2O$ in the 3:2 ratio.

Most interestingly, in the formal oxygen transfer reaction, $\mathbf{4}\mathbf{3}^{2+}(^2H)$ yielded not only the expected product ($[Fe^{III}(OD)$ ($^{5tips3}$tpa)]$^{2+}$, $m/z = 416.3$), but also a product in which the OD group was replaced by OH (i.e. $[Fe^{III}(OH)(^{5tips3}$tpa)]$^{2+}$, $m/z = 415.8$). This observation can be explained by a two-step rebound mechanism. In the first step, hydrogen atom abstraction generates $[Fe^{IV}(OD)(OH)(^{5tips3}$tpa)]$^{2+}$ and a short-lived carbon-centered radical. The radical can be then rebound with either OH or OD from the Fe(OH)(OD) unit, to finally form the corresponding alcohol. The observation of rebound mechanism contrasts with the previously-reported reactivity of iron(IV)-oxo complexes in the gas phase, where the observed oxygen transfer is exclusively due to the epoxidation of C=C double bonds[36]. The opening of the C–H activation pathway in the reaction with 1,3-cyclohexadiene increases the overall abundance of the formal oxygen atom transfer channel over cyclohexene (see Supplementary Table 2). This path occurs in the reaction with 1,3-cyclohexadiene but not with cyclohexene because the latter has a stronger C–H bond (BDE$_{C-H}$ = 74.3 vs 87.0 kcal mol$^{-1}$)[48]. Exactly the same product pattern, identified in all previous reaction channels, was also observed in the reaction of $\mathbf{4}\mathbf{3}^{2+}(^2H)$ with 1,4-cyclohexadiene (Supplementary Fig. 6).

Lastly, we investigated reactions of $\mathbf{4}\mathbf{3}^{2+}$ with aromatic compounds. Reactions with benzene and naphthalene yield addition products followed by water elimination. Furthermore, only in the case of naphthalene, we also observed a single electron transfer reaction, yielding the naphthalene radical cation and a product of single-electron reduced $\mathbf{4}\mathbf{3}^{2+}$[49,50]. Because gas phase reactions only occur when they are exothermic, the electron affinity of $\mathbf{4}\mathbf{3}^{2+}$ must be higher than ionization energy of naphthalene (8.14 eV.)[51]. In turn, this value is higher than the electron affinities of oxoiron(IV) porphyrin cation radicals (cpdI models), which are always lower than 7.5 eV, thus indicating that $\mathbf{4}\mathbf{3}^{2+}$ is a stronger one-electron oxidant than oxoiron(IV) porphyrin cation radicals[50,52]. The addition/water elimination reaction is similar to reactions with cyclohexadiene reactants, but the reaction fully shifts towards final water elimination. The final product regains aromaticity, thereby likely driving the dehydration step kinetically and thermodynamically. This is particularly relevant in the gas phase because the initially formed syn-dihydroxylated product cannot be stabilized by interaction with solvent molecules. On the contrary, we observed the catalytic syn-dihydroxylation of naphthalene by complex **1** and $H_2O_2$ in solution (see supporting information) as also previously observed in reactions with the $[Fe(CH_3CN)_2(tpa)]^{2+}$ complex[53]. We also probed the reaction of $\mathbf{3}^{2+}$ with D$_6$-benzene, and we observed addition followed by HDO elimination with almost 100% selectivity (Fig. 4d). This reaction is highly interesting because these substrates are inert against high-valent Ru and Os oxides and, therefore, show the uniquely powerful oxidation ability of $\mathbf{4}\mathbf{3}^{2+}$.

## Discussion

The current study describes the vibrational and electronic spectroscopic characterization of Fe$^V$(O)(OH) species with a key role in biomimetic oxidations. These Fe$^V$(O)(OH) species have long been proposed to be ultimately responsible for a wide array of oxidations, including enzymatic reactions. However, the inability to accumulate them in solution has thus far prevented their spectroscopic and chemical characterization. Herein, we used gas phase ion spectroscopy methods to address this problem. The electronic and vibrational spectra of these species were finally determined, providing experimental data to unambiguously determine its atomic and electronic structure. Gas-phase reactivity analysis of these well-defined species showed their competence in C–H hydroxylation and syn-dihydroxylation of olefins and arenes. Overall, the data highlights that the particular architecture of the Fe$^V$(O)(OH) species, featuring two reactive ligands in cis-relative positions, translates into singular reactivity properties, unattainable with hemes. For example, high-valent heme iron-oxo complexes consistently epoxidize olefins[4]. However, the current study shows that Fe$^V$(O)(OH) species readily engage in syn-dihydroxylation rather than in epoxidation reactions and, most remarkably, react with arenes. Furthermore, gas phase studies on the hydroxylation of C–H bonds provide direct experimental evidence of a stepwise rebound mechanism, wherein rebound can occur with two different ligands at the Fe center. This behavior differs from that observed in reactions with previously described synthetic Fe$^{IV}$=O complexes, which engage in HAT followed by diffusion of the carbon-centered radical[36,54,55]. Furthermore, this study provides experimental evidence of the rebound of the carbon-centered radical with the two cis-labile ligands at the iron center, which is not possible for hemes because HAT and rebound can only occur at the same oxygen atom[4]. Conversely, in non-heme iron-dependent enzymes and model complexes[56], the presence of labile sites adjacent to the Fe=O moiety enables the transfer of the incipient hydroxyl ligand, or of ligands adjacent to the ferryl. For example, halides, azides and nitrates are transferred in non-heme halogenases[57,58].

Finally, the report shows that gas-phase ion characterization can address questions relevant to catalysis and enzymology, related to highly reactive intermediates, currently unanswerable by other methods.

## Methods

**Generation of intermediate 2**. Initially, 15 μL of a 0.22 M $H_2O_2$ solution in acetonitrile (diluted from 30% in aqueous solution) were directly added over a 2 mL acetonitrile solution of catalyst **1** (0.2 mM). The resulting mixture was cooled to −40 °C with a $CH_3CN/N_2$ (l) bath. At this point, the characteristic purple color of **2** was observed. The solution was kept at −33 °C during the measurements in a two-stage Peltier cooler device.

**Gas phase reactivity**. Mass-spectrometric measurements were performed in a TSQ 7000 quadrupole–octopole–quadrupole spectrometer[43,44]. The ions were transferred to the gas phase using an ESI ion source. Ionization conditions were typically: 6 kV spray voltage, 0 V capillary voltage, 90 V tube lens voltage, 150 °C capillary temperature, 30 psi sheath gas pressure, 300 l h$^{-1}$ auxiliary gas flow. The spray voltage was connected directly to the solution in the vial with a stainless steel wire. The solution was kept at −33 °C and was pumped into the ESI ion source through a 30-cm-long fused-silica capillary with a 100-μm internal diameter by ~1 psi overpressure of nitrogen gas in the vial with the solution. The ions of interest were mass-selected by the first quadrupole and transferred to the octopole equipped with a collision cell; collision gas pressure was determined using a baratron. The collision energy was set to nominally zero, as determined by retarding potential analysis. The products were extracted from the octopole to the second quadrupole, mass-analyzed and detected with a Daly-type detector.

**Helium-tagging infrared/visible photodissociation (IRPD/visPD) spectroscopy**. IRPD/visPD spectra were measured with the ISORI instrument based on the TSQ 7000 platform[32,59,60]. The ions were generated and mass selected exactly as above. The mass-selected ions were transferred via a quadrupole bender and octopole to a cryogenic ion trap operating at 3 K. The ions were trapped with 250 μs helium pulse and formed weakly bound complexes with helium. The trapped ions were irradiated by IR light from an OPO/OPA system or by visible light from a continuum laser wavelength-filtered by acousto-optic tunable filter. After irradiation, all ions were ejected from the trap, mass-analyzed in a second quadrupole and counted by a Daly-type detector. IRPD spectra are plotted as wavenumber-dependent attenuation of the number of helium complexes $(1 - N_i(\nu)/N_{i0})$. The total number of the helium complexes ($N_{i0}$) was obtained in alternative cycles with blocked photon beam. The visPD spectra were corrected by dividing the attenuation by the laser power.

**DFT calculations**. DFT calculations were performed with Gaussian 09[61] at B3LYP-D3/def2TZVP level. All structures were fully optimized and characterized by frequency calculations. The frequencies in IR spectra were scaled by 0.99. Reported energies include zero-point vibrational energy corrections; the molecular coordinates are provided in the Supplementary Note 1 and the relative energies in Supplementary Table 3.

## Data availability

The authors declare that the data supporting the findings of this study are available within the paper and its supplementary information. Further information is also available from the corresponding authors upon reasonable request.

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

## Acknowledgements

M.C. acknowledges the funding from the Spanish Ministry of Economy, Industry and Competitiveness (Ministerio de Economía, Industria y Competitividad – MINECO) (CTQ2015–70795-P and BES-2016–076349), the Catalan DIUE of the Generalitat de Catalunya (2017SGR01378), and the ICREA-Academia award. The project was further funded by the European Research Council (ERC CoG No. 682275). The authors would like to thank Dr. Carlos V. Melo for proofreading the manuscript.

## Author contributions

M.C. and J.R. devised the project, designed the experiments and analyzed the data. M.B. and E.A. performed the experiments and analyzed the data and contributed equally to the manuscript. These authors contributed to the writing of the manuscript. R.N. performed initial DFT calculations and assisted with the IR measurements.

## Additional information

**Competing interests:** The authors declare no competing interests.

