## [Peer Review File · Nature Communications]

Reviewers' comments:

Reviewer #1 (Remarks to the Author):

This manuscript by Roithova and Costas reports the characterization and reactivity of a cis-Fe(V)(O)(OH) complex (3). 3 was characterized using IRPD spectroscopy, such as the Fe=O and Fe-OH stretching vibrations. The reactivity 3 was also demonstrated in the oxidation of olefins and naphthalene. The authors claimed that this study reports the first direct evidence of the Fe(V)(O)(OH) species in non-heme iron catalysis. Indeed, this work should be of interest to those who study non-heme iron enzymes and biomimetic compounds. However, there are already a number of reports on this Fe(V)(O)(OH) and Fe(V)(O)(OC(O)R) (see a recent 2018 JACS paper by one of the authors), I do not think the present work is novel enough to be published in Nature Communications. One reason is that a similar Fe(V)(O)(OC(O)R) species was well characterized spectroscopically (JACS, 2018, 3916) and reactivity studies of the Fe(V)(O)(OH) and Fe(V)(O)(OC(O)R) species have already been reported by Que, Costas, Talsi, etc. So, my question is what is really new in this study. Overall, I do not recommend the publication of this work in Nature Communications.

Comments:

1. While the spin state of other Fe(V)(O) species was spectroscopically determined to be 1/2, the spin state of 3 was proposed to be S=3/2 based on DFT calculations. Why is the spin state of 3 different from that of other Fe(V)(O) species? I suggest that the authors should do DFT calculations for the spin state of Fe(V)(O)(OC(O)R) by replacing the OH ligand with the OC(O)R ligand.
2. It is difficult to follow the reactivity part. How were the reactions of 3 with substrates done? Why do they discuss about rebound mechanism in the cis-dihydroxylation reactions?

Reviewer #2 (Remarks to the Author):

The authors claim to have produced and trapped in the gas phase the FeV(O)(OH)(5tips3tpa)]₂⁺ complex and to have characterized it using gas phase ion spectroscopy. They claim further the identification of key mechanistic events in the C-H bond activation and olefin dihydroxylation and further to be able to distinguish between the rebound mechanism and hydrogen atom abstraction with liberation of a carbon radical with Fe(V) following the former pathway and Fe(IV) the latter pathway. They claim that the gas phase ion spectroscopy results are of value for understanding Rieske dioxygenase reactivity and more generally for understanding the reactivity of highly reactive, hard to isolate and study, reactive intermediates in metal catalyzed oxidation processes.

Overall, I think this is an excellently executed study introducing a novel approach bringing insight into very difficult species to study that merits publication in Nature Communications. In the context that the gas and solution phases are different, I agree with the conclusions of the authors and have no concerns, but I will make a few suggestions at the end of the review.

My enthusiasm arises from these observations:

- The first experimental characterization of cis-FeV(O)(OH) is an important contribution to high valent iron chemistry.
- The characterization is based on beautiful vibrational spectroscopy employing helium-tagging infrared photodissociation (IRPD) spectroscopy with clever labelling experiments and theoretical explorations of the various possible species to create a convincing case for the determined formulation.
- Overall, the quality of the technical work is excellent and the claims are thoroughly bolstered by well presented supplementary information.
- The differential gas phase reactivity of (2)₂⁺ and (3)₂⁺ is interesting in that it shows a

connection between gas and solution phase reactivities as the the authors point out.

- It is always possible that reactions may occur in the electrospray interface that might not occur in solution and the fact that this contingency is detected in the study brings further interest to the work.

Question/Suggestions

- What are "naturally abundant ^{54}Fe -labeled ions ($32+(^{54}\text{Fe})$)"?
- The possibility that long-lived intermediates might undergo intermolecular exchange processes seems not to have been considered. In the cyclohexadiene reaction, H-D exchange processes are attributed to "migrations of protons" in the m/z 464 intermediate. It isn't immediately obvious what the migrations might be. Can we be sure with this technique that water molecules, for example, would not interact with a long-lived intermediate leading to H-D exchanges?
- While the vibrational spectroscopic connection may be absent, the first FeV-oxo complex is extensively characterized by a range of spectroscopic techniques as well as other TAML relatives and its fast expanding solution reactivity remains premier among all iron peroxide activators—probably some more of the leading references in this area could be cited to put this study in better context.

Reviewer #3 (Remarks to the Author):

The paper by Roithová and Costas reports gas-phase generation of an $\text{Fe}=\text{O}(\text{OH})$ species and a combination of vibrational and electronic spectra of the species. I am uncertain about several aspects of the data and conclusions of this paper.

First, it is unclear to me how much of the putative $\text{Fe}=\text{O}(\text{OH})$ species is generated, relative to other species. How does one get the vibrational and electronic spectra if it is not the only species generated? Second, it is unclear to me how confident one can be about OH and OD labeling. A conclusion is drawn about a "two step rebound" mechanism based on observing an OH containing product from an OD starting species. In solution, one would never draw such a conclusion because a proton can be gained in so many ways irrelevant to the mechanism.

At the least these points should be clarified in the paper if the authors want buy-in from more than gas-phase people. If my points are valid concerns, then information on the stability of an OD label in the system and quantitative information on the "yield" of generation of the $\text{FeO}(\text{OH})$ species that is the subject of the paper should be given.

Reviewer #1 (Remarks to the Author):

This manuscript by Roithova and Costas reports the characterization and reactivity of a *cis*-Fe(V)(O)(OH) complex (3). 3 was characterized using IRPD spectroscopy, such as the Fe=O and Fe-OH stretching vibrations. The reactivity 3 was also demonstrated in the oxidation of olefins and naphthalene. The authors claimed that this study reports the first direct evidence of the Fe(V)(O)(OH) species in non-heme iron catalysis. Indeed, this work should be of interest to those who study non-heme iron enzymes and biomimetic compounds. However, there are already a number of reports on this Fe(V)(O)(OH) and Fe(V)(O)(OC(O)R) (see a recent 2018 JACS paper by one of the authors), I do not think the present work is novel enough to be published in Nature Communications. One reason is that a similar Fe(V)(O)(OC(O)R) species was well characterized spectroscopically (JACS, 2018, 3916) and reactivity studies of the Fe(V)(O)(OH) and Fe(V)(O)(OC(O)R) species have already been reported by Que, Costas, Talsi, etc. So, my question is what is really new in this study. Overall, I do not recommend the publication of this work in Nature Communications.

Authors reply: Oxoiron(V) species are extraordinarily rare, despite they are proposed as intermediates in a number of catalytic cycles. Examples shown in Table 1 are the only examples. As noticed by the reviewer, we described recently the first spectroscopic characterization of the Fe(V)(O)(OC(O)R) intermediate and showed that it is catalytically relevant. The Fe(V)(O)(OH) species may be a priori considered similar but it has fundamental differences and singularities that justify its unique interest;

- a) The Fe(V)(O)(OH) species has been proposed since 1998 as the species responsible for *syn*-dihydroxylation in biomimetic catalysis and most importantly, it is also proposed as the active species in Naphthalene dioxygenase, an enzyme of the family of Rieske Oxygenases. Therefore, these are species of relevance in oxidation catalysis and in enzymology. However, it has never been characterized. This work provides their first spectroscopic characterization. The compound is particularly remarkable because it represents the only Fe(V) species that nowadays is proposed to have a biological role. Other formal Fe(V) species such as Cpdl in P450 are instead oxoiron(IV) species with a porphyrin ligand radical. In addition, the related Fe(V)(O)(OC(O)R) species, which have relevance in bioinspired catalysis, has never been considered of relevance in biological systems.
- b) The chemistry exhibited by the two species is substantially different. While Fe(V)(O)(OC(O)R) performs epoxidation, and overall mirrors the chemistry of hemes, Fe(V)(O)(OH) has been proposed to be the species responsible for *syn*-dihydroxylation of arenes and alkenes. Our work shows that they indeed perform *syn*-dihydroxylation of these substrates. Furthermore, the C-H hydroxylation mechanism exhibited by the Fe(V)(O)(OH) species is singular, because after the initial hydrogen atom transfer by the Fe=O, the hydroxyl rebound is shown to occur with the two Fe-OH ligands. For any other Fe=O species (heme and non-heme), hydrogen abstraction and rebound occurs at the same oxygen atom (originally Fe=O). This is relevant because this feature (rebound with the ligand adjacent to the terminal oxo) has been for long proposed as a unique feature of non-heme iron-dependent oxygenases and halogenases.
- c) The electronic structure of the two Fe(V) species is also different. The Fe(V)(O)(OH) species is a $S = 3/2$ system, as it has been predicted computationally for all the systems where such intermediate has been proposed to participate. Instead, all the spectroscopically characterized examples of Fe(V) complexes (included the Fe(V)(O)(OC(O)R)) are $S = 1/2$.

Finally, the methodology employed in the current work is envisioned to have general interest for people working on reaction mechanisms because it permits to access spectroscopic and chemical characterization of transient species that do not accumulate in solution and elude characterization by more conventional spectroscopic and kinetic methods.

Overall, we believe that the work collects very significant elements of novelty and scientific relevance.

Comments:

1. While the spin state of other Fe(V)(O) species was spectroscopically determined to be 1/2, the spin state of 3 was proposed to be S=3/2 based on DFT calculations. Why is the spin state of 3 different from that of other Fe(V)(O) species? I suggest that the authors should do DFT calculations for the spin state of Fe(V)(O)(OC(O)R) by replacing the OH ligand with the OC(O)R ligand.

Authors reply: We have performed the suggested DFT calculations and they are consistent with our conclusions. The ground state of Fe(V)(O)(OC(O)Me) is always S=1/2 and is higher in energy than its Fe(III)(OOC(O)Me (ferric peracetate "closed form"). Hence, the preference for the oxohydroxo geometry over the hydroperoxo coordination results in stabilizing the S=3/2 state. We have added a detailed explanation and the new calculations to the Supplementary Material.

2. It is difficult to follow the reactivity part. How were the reactions of 3 with substrates done? Why do they discuss about rebound mechanism in the cis-dihydroxylation reactions?

Authors reply: We have rewritten the reactivity part to make it clear. We hope that we removed the confusion.

Reviewer #2 (Remarks to the Author):

The authors claim to have produced and trapped in the gas phase the FeV(O)(OH)(5tips3tpa)]²⁺ complex and to have characterized it using gas phase ion spectroscopy. They claim further the identification of key mechanistic events in the C–H bond activation and olefin dihydroxylation and further to be able to distinguish between the rebound mechanism and hydrogen atom abstraction with liberation of a carbon radical with Fe(V) following the former pathway and Fe(IV) the latter pathway. They claim that the gas phase ion spectroscopy results are of value for understanding Rieske dioxygenase reactivity and more generally for understanding the reactivity of highly reactive, hard to isolate and study, reactive intermediates in metal catalyzed oxidation processes.

Overall, I think this is an excellently executed study introducing a novel approach bringing insight into very difficult species to study that merits publication in Nature Communications. In the context that the gas and solution phases are different, I agree with the conclusions of the authors and have no concerns, but I will make a few suggestions at the end of the review.

My enthusiasm arises from these observations:

- The first experimental characterization of cis-FeV(O)(OH) is an important contribution to high valent iron chemistry.
- The characterization is based on beautiful vibrational spectroscopy employing helium-tagging

infrared photodissociation (IRPD) spectroscopy with clever labelling experiments and theoretical explorations of the various possible species to create a convincing case for the determined formulation.

- Overall, the quality of the technical work is excellent and the claims are thoroughly bolstered by well presented supplementary information.
- The differential gas phase reactivity of (2)2+ and (3)2+ is interesting in that it shows a connection between gas and solution phase reactivities as the authors point out.
- It is always possible that reactions may occur in the electrospray interface that might not occur in solution and the fact that this contingency is detected in the study brings further interest to the work.

Question/Suggestions

- What are "naturally abundant ^{54}Fe -labeled ions (32+(^{54}Fe))"?

Authors reply: Iron has several naturally occurring isotopes that are always present (unless the sample is isotopically purified). The most abundant isotope is ^{56}Fe (~92%), ^{54}Fe is present in almost 6%. We have change the word "abundant" to "occurring".

- The possibility that long-lived intermediates might undergo intermolecular exchange processes seems not to have been considered. In the cyclohexadiene reaction, H-D exchange processes are attributed to "migrations of protons" in the m/z 464 intermediate. It isn't immediately obvious what the migrations might be. Can we be sure with this technique that water molecules, for example, would not interact with a long-lived intermediate leading to H-D exchanges?

Authors reply: We are working in the gas phase with isolated individual ions that are interacting each with just one molecule of reactant. There is no involvement of other molecules. We have rephrased the description of the experiments to make is easier to follow.

- While the vibrational spectroscopic connection may be absent, the first FeV-oxo complex is extensively characterized by a range of spectroscopic techniques as well as other TAML relatives and its fast expanding solution reactivity remains premier among all iron peroxide activators—probably some more of the leading references in this area could be cited to put this study in better context.

Authors reply: Two recent references, including a review, covering FeTAML's systems have been included

Reviewer #3 (Remarks to the Author):

The paper by Roithová and Costas reports gas-phase generation of an Fe=O(OH) species and a combination of vibrational and electronic spectra of the species. I am uncertain about several aspects of the data and conclusions of this paper.

First, it is unclear to me how much of the putative Fe=O(OH) species is generated, relative to other

species. How does one get the vibrational and electronic spectra if it is not the only species generated?

Authors reply: Using mass spectrometry approach, it is not possible to obtain concentration of the species in solution. We have also pointed out that the complex that we investigate in the gas phase is most probably formed by the rearrangement of the hydrogenperoxo complex. It was shown previously that iron(V) complex in solution is formed by a different mechanism.

The power of our approach consists in combination of mass spectrometry and IR/UV/vis spectroscopy. Hence, we mass-select the complexes of the interest (we can separate ions that have miniscule concentration) and then determine their IR and UV-vis spectra (and reactivity). Therefore we can individually study many different complexes that can be formed from the given reaction mixture.

Second, it is unclear to me how confident one can be about OH and OD labeling. A conclusion is drawn about a “two step rebound” mechanism based on observing an OH containing product from an OD starting species. In solution, one would never draw such a conclusion because a proton can be gained in so many ways irrelevant to the mechanism.

Authors reply: We are working with individual mass-selected ions that interact each with just one molecule of reactant (no other molecules are involved). Therefore, we know exactly how many H and D atoms we have in the reactant complex and we can follow their distribution in products. We have rephrased this part of the manuscript to make it more clear.

At the least these points should be clarified in the paper if the authors want buy-in from more than gas-phase people. If my points are valid concerns, then information on the stability of an OD label in the system and quantitative information on the “yield” of generation of the FeO(OH) species that is the subject of the paper should be given.

Authors reply: We hope that we made the points of concern more clear to avoid misunderstandings.